# The Effect of Financial Market Factors on House Prices: An Expected Utility Three-Asset Approach

**Yehui Wang** [1] , **Jianxu Liu** [1,2,*] , **Zhaolin Qiu** [1] **and Songsak Sriboonchitta** [2]

1   Department of Economics, Shandong University of Finance and Economics, Jinan 250014, China; yehui.wang@hotmail.com (Y.W.); qiuzhaolin1985@126.com (Z.Q.)
2   Center of Excellence in Econometrics, Faculty of Economics, Chiang Mai University, Chiang Mai 50200, Thailand; songsakecon@gmail.com
*   Correspondence: liujianxu1984@163.com

**Abstract:** This study aimed to theoretically identify the impact factors of the financial market on house prices. Developed upon the two-asset model and with the consideration of risky financial assets, our three-asset model reveals a new derivation of house prices. Compared with the two-asset model, the newly emerged term is similar to the Sharpe $\beta$; therefore, it is a risk premium term. Based on China's 2001–2018 panel data, theoretical derivations are examined. However, the short-term effect of this risk term on house prices is practically small. Given the nonlinear pattern, the long-term effect of the risk term is checked by repeated stochastic simulation. The results imply the following: (i) real house prices are nonlinearly affected by three financial market factors, namely, the expected financial market return, financial market volatility, and the correlation between housing and financial markets; (ii) the correlation determines the signs and the significance of the effects of the other two factors; and (iii) the naturally changed correlation causes periodic house price fluctuations. Therefore, to stabilize real house prices, it is recommended that the government control the money flow between the two markets.

**Keywords:** house price; risk premium; financial market factors

**MSC:** 91B02

## 1. Introduction

House prices have been sharply inflated internationally in recent decades. Most people believe that the financial market plays an important role in housing market appreciation. It seems that house prices always increase in boom periods and decrease during financial crises. However, related empirical works have not supported this view. Muellbauer and Murphy [2] pointed out that house price changes are related to financial market liberalization in the UK. Green [3] showed that the impact of stock prices on house prices is high in Northern California but low in Southern California in the US. Lee et al. [4] focused on the relationship between Australian housing and stock markets and found that stock market prices were affected by house prices before the recent GFC but affected house price after the crisis. Kakes and Van Den End [5] found that house prices are significantly affected by stock market prices in the Netherlands. Takala and Pere [6] found that house prices are Granger-caused by stock prices in Finland. Ibrahim [7] applied the same method but found that there was no significant Granger-causality between housing and financial markets in Thailand. All of these related empirical works have suggested the following: (i) the impact factors of the financial market on house prices are unclear; (ii) their influence mechanisms are unclear; and (iii) the effects of the financial market factors on house prices should be much more complicated than we believed. Therefore, related theoretical work on the effects of financial market factors on house prices is necessary.

Housing is different from most financial assets since houses not only have investment demand but also have consumption demand, usually known as the dual property [8]. Houses provide housing services at every moment for people living in the house. Due to the consumption demand, house prices are affected by several economic factors, such as household income, family wealth, and construction costs [8]. These impacts are usually theoretically derived by consumption type models, especially the housing life-cycle model [9]. In contrast, housing is a type of asset that can be selected and invested in by investors. Based on the determinants of housing investment demand, the prices of houses are affected by many financial factors, such as (mortgage) interest rates, expected financial market return, the variances (or the standard deviations) of both housing and financial market returns, and the covariance (or the correlation) between the returns of the two markets [10–12]. These factors are derived from financial theories, for example, the Consumption-based Capital Asset Pricing Model (C-CAPM) and Housing CAPM [10,13–16]. However, although some studies [17] have nested the economic and financial factors in their empirical analyses and found that both of these factors are significant, it remains controversial whether it is appropriate to assume housing to be a type of pure financial assets and then to analyse house prices using financial theories [18].

In economics, the housing life-cycle model [9] is popularly used in analysing housing issues since houses are clearly defined as both consumption and investment products in the model. Despite the standard housing life-cycle model allowing for the consideration of the dual property of housing, as a result of the underlying assumption of certainty, it still has no position in risk issues. The expected utility two-asset housing life-cycle model (hereinafter, two-asset model) that [1] developed had a theoretical framework based on the standard model, in which (i) housing is assumed to be risky, and thus future house prices are assumed to be uncertain, like a normal distribution; and (ii) households maximize their expected lifetime utility. The results showed that the volatility of housing capital return has significant and non-negligible negative impacts on real house prices, particularly in long-run forecasting. The missing consideration of the effect of the volatility will cause over-estimation. However, the two-asset model remains incomplete since to assume houses are the only risky asset in the market is not sufficient. The effects of financial market factors on housing variables are also worth being analysed. In addition, the time-varying [19] and regional differed [20] distributions of house prices as well as the volatilises are also possibly caused by the factors from financial market or related government policies. These issues will be discussed in the future studies.

In light of these discussions, there have been three main gaps recently in this area. First, it is necessary to have a theoretical base in which, at least, risky housing and risky financial assets are jointly considered. Second, given the consumption-based model, households' choices of risky assets, including housing and risky financial assets, should be derived and compared with those implied in financial theories. Further, the reasons why the results are consistent or different should be discussed. Third, the theoretical derivation should reflect the pattern of how financial market factors affect house prices, which would be particularly helpful for related empirical works.

Therefore, this article is organized as follows. Section 2 is the theoretical section, which provides the details of the three-asset model, including related theoretical derivations. Section 3 is the empirical section that practically estimates the short-term effect of housing market risk premiums on house prices. In Section 4, based on reasonable assumptions, the dynamic impact of the cyclically fluctuated correlation with real house prices is simulated. Section 5 provides the conclusion and some political suggestions. References and appendices are provided at the end.

## 2. Theory

In this section, the expected utility three-asset housing life-cycle model (hereinafter, three-asset model) is introduced. In related work [1], housing was assumed to be risky, and the housing market risk premium appeared in the housing user cost of capital (UCC), as well as the house price derivation. However, when risk issues are considered, the as-

sumption of merely unique risk housing assets is insufficient. At least, all other financial assets should be categorized into two groups: risk-free and risky. Therefore, the theoretical structure and some details of the three-asset model are first provided, and then the general solution of the model is derived. Under two specific utilities, the Constant Absolute Risk Aversion (CARA) and the Constant Relative Risk Aversion (CRRA) utilities, special solutions are derived. Because of the joint consideration of two risk assets, the potential households' portfolio decisions are derived and discussed.

### 2.1. The Expected Utility Three-Asset Housing Life-Cycle Model

Based on the standard inter-temporal housing life-cycle model [9] and the expected utility two-asset housing life-cycle model [1], household temporal utility is determined by non-housing consumption ($C$) and the used volume of housing services, which is proportional to the housing stock ($H$) as $\mu[H(t), C(t)]$. For "now," household's utility $\mu[H(0), C(0)]$ is certain, but all future $t > 0$ utilities are uncertain. Inter-temporally, given an assumed constant real discount rate ($r$), the present value of the lifetime utility is described for an infinite horizon discrete time ($t$) as $\sum_0^\infty \frac{1}{(1+r)^t} \mu[H(t), C(t)]$ or continuous time ($t$) as $\int_0^\infty e^{-rt} \mu[H(t), C(t)] dt$. A rational household now will make decisions to maximize the expectation of the continuous type present value of its uncertain lifetime utility ($U$), given by Equation (1).

$$U = E\left\{ \int_0^\infty e^{-rt} \mu[H(t), C(t)] dt \right\} \tag{1}$$

Households' expected lifetime utility is maximized with respect to budget constraint (2) and technical constraints (3)–(5), which describe changes in the three real asset stocks (risky housing, risk-free assets, and risky financial assets, respectively) over time. Generally, based on the expected returns of these three assets, households will make an investment decision on the three assets for the purpose of maximizing their expected lifetime utility. In addition, for the purpose of simplicity, the issues of taxes [21,22], transaction costs [23], liquidity [24], and restrictions on lending [25] are removed.

$$p(t)C(t) + p_H(t)X(t) + p(t)S_f(t) + p_A(t)S_A(t) = Y(t) + i(t)p(t)A_f(t) + i_d(t)p_A(t)A_A(t) \tag{2}$$

$$\dot{H}(t) = X(t) - \delta(t)H(t) \tag{3}$$

$$\dot{A}_f(t) = S_f(t) - \pi(t)A_f(t) \tag{4}$$

$$\dot{A}_A(t) = S_A(t) - \pi(t)A_A(t) \tag{5}$$

where
$p(t)$ = price level/Consumer Price Index(CPI);
$C(t)$ = quantity of non-housing consumption;
$p_H(t)$ = nominal purchase price of dwellings;
$X(t)$ = quantity of new purchases of dwellings;
$S_f(t)$ = quantity of savings net of real new loans volume of risk-free assets;
$p_A(t)$ = nominal purchase price of risky financial assets;
$S_A(t)$ = quantity of savings net of real new loans volume of risky assets;
$Y(t)$ = nominal disposable income;
$i(t)$ = market risk-free/interest rate;
$A_f(t)$ = quantity of net non-housing risk-free assets;
$i_d(t)$ = dividend yield of the financial risky asset;
$A_A(t)$ = quantity of net non-housing risky assets;
$\delta(t)$ = depreciation rate on housing;
$\pi(t)$ = general inflation rate; and
$\dot{x} \equiv dx(t)/dt$, denotes the time derivative for any variable x(t).

Budget constraint (2) implies the equivalence between nominal terms. By dividing the price level (*p*) for both sides, Equation (6) indicates the equivalence between real terms. It shows that the flow of net expenditures on non-housing consumption, housing, risk-free savings, and savings on risky financial assets is equal to the net earnings from labour, risk-free investments, and risky financial investments. Now, housing acts as both consumption and investment since housing stock (service) directly determines household' instantaneous utility and, at the same time, inflates in a similar pattern to financial assets. Additionally, some have asked why there is no housing rent/yield in the budget constraint. It is because, housing rent, in principle paid by all people, is equal to the housing rental income obtained by all households, so they are cancelled on both sides of Equations (2) and (6).

$$C(t) + g_H(t)X(t) + S_f(t) + g_A(t)S_A(t) = RY(t) + i(t)A_f(t) + i_d(t)g_A(t)A_A(t) \quad (6)$$

where
$g_H(t)$ = real purchase price of dwellings;
$g_A(t)$ = real purchase price of the financial risky assets; and
$RY(t)$ = real disposable income.

Mathematically, to maximize lifetime utility (1) subjected to (3)–(6), a Lagrange equation is structured. From the first-order conditions (see Appendix A.1), we achieved three Equations (7)–(9).

$$\frac{\partial \mathcal{L}(t)}{\partial H(t)} : E[\mu_H(t)] = g_H(t)\left\{ E[\mu_C(t)]\delta(t) - E[r_H(t)\mu_C(t)] - e^{rt}E\left[\frac{d}{dt}[e^{-rt}\mu_C(t)]\right] \right\} \quad (7)$$

$$\frac{\partial \mathcal{L}(t)}{\partial A_f(t)} : [i(t) - \pi(t)]E[\mu_C(t)] = -e^{rt}E\left[\frac{d}{dt}[e^{-rt}\mu_C(t)]\right] \quad (8)$$

$$\frac{\partial \mathcal{L}(t)}{\partial A_A(t)} : E\left[[r_A(t) + i_d(t) - \pi(t)]\mu_C(t)\right] = -e^{rt}E\left[\frac{d}{dt}[e^{-rt}\mu_C(t)]\right] \quad (9)$$

where
$\mu_H(t) = \partial\mu[H(t), C(t)]/\partial H(t)$, the (uncertain) marginal utility of housing;
$\mu_C(t) = \partial\mu[H(t), C(t)]/\partial C(t)$, the (uncertain) marginal utility of consumption;
$r_H(t) = \dot{g}_H(t)/g_H(t)$, the (uncertain) capital return of housing; and
$r_A(t) = \dot{g}_A(t)/g_A(t)$, the (uncertain) capital return of risky financial asset.

Since $r_H$ and $r_A$ are functions of future states (e.g., $t + dt$), returns on risky assets are uncertain. Similarly, the marginal utilities of both housing and non-housing consumption are uncertain since they are functions of $r_H$ and $r_A$, respectively. Due to the same term $-e^{rt}E\left[\frac{d}{dt}[e^{-rt}\mu_C(t)]\right]$ in all three of these equations, shown in Appendix A.2, the relationship between risk-free and risky financial assets is derived in (10) through the combination of (8) and (9); similarly, the relationship between housing and risk-free assets is derived in (11) through the combination of (7) and (8).

$$\frac{E\left[r_{AD}(t)\mu_C(t)\right]}{E[\mu_C(t)]} = i(t) - \pi(t) \quad (10)$$

$$\frac{E[\mu_H(t)]}{E[\mu_C(t)]} = [i(t) - \pi(t) + \delta(t) - E(r_H(t)) + \tau(t)]g_H(t) \quad (11)$$

where
$E[\mu_H(t)] = \partial E[\mu[H(t), C(t)]]/\partial H(t)$, the expected marginal utility of housing;
$E[\mu_C(t)] = \partial E[\mu[H(t), C(t)]]/\partial C(t)$, the expected marginal utility of consumption;
$r_{AD}(t) = r_A(t) + i_d(t) - \pi(t)$, the (uncertain) total return of risky financial assets; and

$\tau(t) = -Cov[r_H(t), \mu_C(t)]/E[\mu_C(t)]$, the housing market risk premium.

Equation (10) is identical to the key derivation of C-CAPM [13], and Equation (11) is identical to the key derivation indicated in the two-asset model [1], where a housing market risk premium term, $\tau$, appeared in the house price derivation compared to the standard derivation [9].

The left-hand side of (11) is the expected marginal rate of substitution between $H$ and $C$, meaning that households must cut down on unit housing service if they want to obtain $R$ units of non-housing goods as compensation; therefore, $R$ is the imputed real housing rental price. The term in $[\dots]$ on the right-hand side of (11) is the widely used standard definition of the real housing user cost of capital (UCC). All of these factors are summarized by Equation (12). This equation can be also rewritten as $g(t) = R(t)/UCC(t)$, the well-known arbitrage condition.

$$R(t) = E[MRS_{H,C}(t)] = \frac{E[\mu_H(t)]}{E[\mu_C(t)]} = UCC(t)g_H(t) \tag{12}$$

where
$R(t)$ = the imputed real housing rental price;
$E[MRS_{H,C}(t)]$ = the expected marginal rate of substitution between $H$ and $C$; and
$UCC(t) = i(t) - \pi(t) + \delta(t) - E[r_H(t)] + \tau(t)$, the user cost of capital.

Equations (10)–(12), particularly (11), are the general solutions of our key interest. However, since they are too general, we must try very hard to use them to estimate housing market risk premiums in empirical works.

### 2.2. Special Solutions under the Specific Utility Functions

The last section theoretically derived and described the general solution of house prices and housing market risk premiums. In principle, the solution can be adopted for all countries, which is also why a general solution is too abstract. To be able to derive intuitive results of $\tau$ for a specific region, specific utility functions and distribution of the returns are needed. For the purpose of showing the calculation process, the distribution of the risky returns is here simplistically assumed to be joint normal as an example, given by (13). Similarly, the Constant Absolute Risk Aversion (CARA) and Constant Relative Risk Aversion (CRRA) utilities [26] given by (14) and (15), respectively, are employed.

$$\begin{bmatrix} r_H(t) \\ r_{AD}(t) \end{bmatrix} \sim N\left( \begin{bmatrix} r_H^e(t) \\ r_{AD}^e(t) \end{bmatrix}, \begin{bmatrix} \sigma_H^2(t) & \sigma_{AH}(t) \\ \sigma_{AH}(t) & \sigma_{AD}^2(t) \end{bmatrix} \right) \tag{13}$$

$$\mu[H(t), C(t)] = -exp[-\varphi H(t)] - exp[-\varphi C(t)] \tag{14}$$

$$\mu[H(t), C(t)] = \frac{H(t)^{(1-\theta)}}{1-\theta} + \frac{C(t)^{(1-\theta)}}{1-\theta} \tag{15}$$

where
$\varphi$ = the parameter of CARA;
$\theta$ = the parameter of CRRA;
$\sigma_{AH} = \rho \times \sigma_{AD} \times \sigma_H$, the covariance between $r_{AD}$ and $r_H$; and
$\rho$ = the correlation between $r_{AD}$ and $r_H$.

Since there are two utility functions, Appendices A.3 and A.4 separately show the process of deriving housing market risk under the CARA and CRRA utilities, and the special solutions are given by (16)–(18) and (19)–(21), respectively.

$$\tau_{CARA} = \tau_{1,CARA} + \tau_{2,CARA} \tag{16}$$

$$\tau_{1,CARA}(t) = [r_{AD}(t) - (i(t) - \pi(t))] \times [\rho(t)\sigma_H(t)/\sigma_{AD}(t)] \tag{17}$$

$$\tau_{2,CARA}(t) = \varphi g_H(t)H(t)(1-\rho^2(t))\sigma_H^2(t) \tag{18}$$

$$\tau_{CRRA} = \tau_{1,CRRA} + \tau_{2,CRRA} \tag{19}$$

$$\tau_{1,CRRA}(t) = [r_{AD}(t) - (i(t) - \pi(t))] \times [\rho(t)\sigma_H(t)/\sigma_{AD}(t)] \tag{20}$$

$$\begin{aligned}
\tau_{2,CRRA}(t) &= \frac{-\theta C^e(t)^{-\theta-1}}{E[\mu_C(t)]} g_H(t) H(t)(1-\rho^2(t))\sigma_H^2(t) \\
&\approx [\theta/C^e(t)] g_H(t) H(t)(1-\rho^2(t))\sigma_H^2(t)
\end{aligned} \tag{21}$$

First, both $\tau_{CARA}$ and $\tau_{CRRA}$ consist of two parts: $\tau_1$ and $\tau_2$. Second, the $\tau_1$ of the two cases is identical. When risky financial assets are regarded as the whole market, and housing is regarded as a particular asset in this market, $r_{AD} - (i - \pi)$ becomes the expected exceed return of the financial market, and $\rho \times \sigma_H/\sigma_{AD}$ (or $\sigma_{AH}/\sigma_{AD}^2$) becomes the systematic risk of housing in finance, also known as Sharpe $\beta$ [27]. Therefore, $\tau_1$ is the housing market risk premium affected by the financial market elements. Third, $\tau_2$ is the housing market risk directly caused by the uncertainty of the housing market yield, which has been discussed in the two-asset model [1].

In light of these considerations, $\tau_1$ is the main focus in this article. It is determined by three financial market factors: the expected exceed return on the risky financial asset ($r_{AD} - (i - \pi)$), the standard deviation ($\sigma_{AD}$), and the correlation ($\rho$). In principle, the former two are positive, and the latter is in the interval $[-1, 1]$. The correlation is key in determining the effects of other financial market elements. When $\rho > 0$, relatively higher expected returns or lower volatility of the financial market indicates lower real house prices; when $\rho < 0$, real house prices are positively determined by the expected return and negatively determined by the volatility. Simply speaking, when the two markets are strongly "substituted," financial market flourishing causes lower house prices; when the two markets are strongly "compensated," financial market flourishing indicates higher house prices; and when the correlation is relatively tiny, the effects of financial market factors on housing market risk premiums, as well as house prices, can be neglected.

Our $\tau_2$ derivations (18) and (21) are similar to Equations (18) and (19) shown in the two-asset model [1]. The effects of risk aversion level ($\varphi$ and $\theta/C$), real housing wealth ($g_H H$), and the volatility of housing capital return ($\sigma_H^2$) have been discussed and analysed [1]. The only difference is that term $(1 - \rho^2)$ newly appears in our $\tau_2$ in both CARA and CRRA cases. Whether positive or negative, a strong correlation leads to $(1 - \rho^2) \longrightarrow 0$, implying that the housing market risk premium caused by the uncertainty of the housing market ($\tau_2$) will be eliminated when the two markets reflect strong correlations in their market returns because, when the two markets have no interaction, the volatility of the housing capital return ($\sigma_H^2$) can fully capture the uncertainty of the housing market. In contrast, when the two markets are related, the correlated part ($\sigma_H^2\rho^2$) will decrease from the volatility since this risk is dispersed by households' asset portfolios.

Therefore, the special solutions indicate that financial market factors determine the housing market risk premium and thus affect real house prices. The correlation is the key in both of $\tau_1$ and $\tau_2$. A tiny correlation between the two markets means tiny financial market impacts but significant $\tau_2$. However, despite significant correlations being able to hide $\tau_2$, the effects of financial market factors on house prices are significant. Hence, we can see that both de- and over-financialization are bad for housing markets. To answer the question of the appropriate level of financialization of the housing market, the risk dispersion mechanism through market correlation must be further studied.

### 2.3. Portfolios between Housing and Risky Financial Assets

Derivation $\tau_1$ reflects a highly similar structure to the systematic risk derivation or the Sharpe $\beta$. Remember that both CAPM and its predecessor, the mean-variance model [28], are concentrated on portfolios among risky assets and then derive the price/return to risk premium relation. Here, we undergo a reversed process since we have the price/return to risk premium relation of housing ($\tau_1$) and want to know the portfolio solution.

As shown in Appendix A.5, conditional on the CARA utility function, the optimal ratio of risky financial wealth to housing wealth is derived and given by (22); similarly,

conditional on the CRRA utility function, the derivation of the optimal ratio is also (22). Appendix A.6 provides the main body of the mean-variance model [28]. When housing and risky financial assets are the two specific assets A and B, its derivation of the optimal ratio is identical to (22), suggesting that our three-asset model has in-built households optimal portfolio decisions.

$$
\begin{aligned}
k^*(t) &= \frac{g_A(t)A_A(t)}{g_H(t)H(t)} \\
&= \frac{\left[r^e_{AD}(t) - [i(t) - \pi(t)]\right]\sigma^2_H(t) + \left[r^e_{HT}(t) - [i(t) - \pi(t)]\right]\sigma_{AH}(t)}{\left[r^e_{HT}(t) - [i(t) - \pi(t)]\right]\sigma^2_{AD}(t) + \left[r^e_{AD}(t) - [i(t) - \pi(t)]\right]\sigma_{AH}(t)}
\end{aligned}
\tag{22}
$$

where $r^e_{HT}(t) = r^e_H(t) + R(t)/g(t) - \delta(t)$, the expected total housing return.

One thing that must be mentioned is that, despite the variance of risky financial assets considering both the variances of capital and dividend yields, the variance of housing is only the variance of the capital yield without housing rental yield because the dividends of financial instruments are usually uncertain, but, practically, almost all housing rents are agreed and paid before tenants move in. In light of this difference, the housing imputed rent is assumed to be certain and thus has no risk.

All of the above results suggest that, when households make their optimal decisions to maximize their expected lifetime utility, their decisions are not only optimal in consumption–investment balance [9] but also include instantaneous optimal portfolio decisions in investing in housing and risky financial assets.

## 3. Empirical Section

The main purpose of this section is to check whether housing market risk premiums, especially $\tau_1$, are significant or not. As given in Wang et al. [1], Chinese data are employed for three reasons. First, the China Statistic Office (CSO) provides both monthly and annual house prices. Thus, the volatility of the latter (annual variance) could be calculated from the former. Second, provincial data are available, and therefore, empirical relationships can be analysed by panel data regressions, which are more robust than single equation time series regressions. Third, the housing price index provided by CSO is the price per square meter. Compared to the price per house index, it is less likely that the prices have strong regional heterogeneity.

### 3.1. Data and Testing

The complete data set is annual panel data including China's 30 provinces except for Tibet, Taiwan, Hong Kong, and Macao for the period of 2001–2018. Details of the variables are listed in Table 1. The interest rate ($i$) is proxied by the base interest rate on RMB (CNY, the Chinese currency) loans provided by China Financial Statistical Yearbook; the market price of risky financial assets of China is approximated by the Shanghai Securities Composite Index (SSEC). Except for these two factors, all of the other variables are obtained from China Statistical Yearbook. As shown in Table 1, the annual expected returns, standard deviations, and correlations are calculated through the monthly growths/returns of house prices and SSEC. Since there are no high-quality housing stock data, as discussed in the two-asset model [1], the flow of real housing wealth ($HE/CPI$) is selected to be the approximation of $g_H(t)H(t)$. Furthermore, for the purpose of avoiding confusion, we only focus on the CRRA case in the empirical section.

**Table 1.** Data Description.

| Name | Variable | Label/Measurement | Mean | S.D. |
|------|----------|-------------------|------|------|
| Monthly: | | | | |
| *MHE* | monthly *HE* | 100 million yuan | | |
| *MHS* | monthly *HS* | 10 thousand m$^2$ | | |
| *MHP* | monthly *HP* | $= HE/HS \times 10,000$ | | |
| *MPA* | SSEC | – | | |
| *mrh* | *MHP* monthly growth | $mrh_m = ln(MHP_m) - ln(MHP_{m-1})$ | | |
| *mra* | *MPA* monthly growth | $mra_m = ln(MPA_m) - ln(MPA_{m-1})$ | | |
| Annual: | | | | |
| *HE* | housing expenditure | 100 million yuan | 1592.1 | 2194.1 |
| *HS* | housing space sold | 10 thousand m$^2$ | 2862.4 | 2765.5 |
| *HP* | house price (yuan/m$^2$) | $= HE/HS \times 10,000$ | 4566.0 | 3730.9 |
| *Y* | disposable income | yuan/person | 18,218.8 | 10,332.2 |
| *CPI* | consumer price index | 2000 = 100 | 123.1 | 17.72 |
| *CON* | final consumption | 100 million yuan | 6136.5 | 6342.3 |
| *G* | real house price | $G = HP/CPI$ | 35.953 | 27.701 |
| *lng* | logarithm form | $lng = ln(G)$ | | |
| *RY* | real income | $RY = Y/CPI$ | 142.028 | 68.680 |
| *lnry* | logarithm form | $lnry = ln(RY)$ | | |
| *i* | (loan) interest rate | – | 0.053 | 0.0056 |
| *rh* | nominal return | $rh_{-1} = r_H + \pi$ | 0.095 | 0.093 |
| *tau*1 | $\tau_1$ | Equation (20) | −0.0036 | 0.084 |
| *tau*2 | $\tau_{2,CRRA}$ | Equation (21) | 0.035 | 0.100 |
| *rhe* | expected housing yield | $rhe_t = (1/12)\sum_{m=Jan}^{Dec} mrh_{t,m} \times 12$ | | |
| *rae* | expected financial yield | $rae_t = (1/12)\sum_{m=Jan}^{Dec} mra_{t,m} \times 12$ | | |
| *SDH* | S.D. of $r_H$ | $\sigma_{H,t} = \sqrt{(1/12)\sum_{m=Jan}^{Dec}(mrh_{t,m} - rhe_t)^2}$ | | |
| *SDA* | S.D. of $r_{AD}$ | $\sigma_{AD,t} = \sqrt{(1/12)\sum_{m=Jan}^{Dec}(mra_{t,m} - rae_t)^2}$ | | |
| *corr* | correlation | $\rho_t = \sigma_{[H,AD],t}/(\sigma_{H,t}\sigma_{AD,t})$ | | |

Notes: (i) The time subscripts *t* and *m* are year and month, respectively; (ii) SSEC = Shanghai Securities Composite Index; (iii) "×12" in *rhe* and *rae* indicates annualization; (iv) $\sigma_{[H,AD]}$ is the covariance between *mrh* and *mra*.

In empirical analysis, the stationarity of variables determines the panel model selection. The stationarity of variables in the panel structure is usually tested by LLC, IPS, ADF-Fisher, and Hadri LM stationary tests. The stationarity of *lnry*, *lng*, and *i* were already tested by Wang et al. [1], and the results suggested that (log) real house prices and (log) real income contain unit-roots in all panels, but the interest rate (*i*) is stationary in all panels. Similarly, the stationarity of $\tau_1$ and $\tau_{2,CRRA}$ is tested by all four of these stationary tests, as shown in Table 2. The results support that both $\tau_1$ and $\tau_{2,CRRA}$ are stationary for all panels.

**Table 2.** Panel Data Unit-Root/Stationary Test (*p*-value).

| Test | *tau*1 | *tau*2 |
|------|--------|--------|
| Levin-Lin-Chu (LLC) test | <0.001 | <0.001 |
| Im-Pesaran-Shin (IPS) test | <0.001 | <0.001 |
| ADF-Fisher unit-root test (lag = 2) | <0.001 | <0.001 |
| Hadri LM stationary test | 0.680 | 0.726 |

*3.2. Regression Analysis*

The empirically used equations are slightly different compared to the theoretical derivations because: (i) the imposed housing rent (*R*) is unavailable; and (ii) the logarithm of user cost of capital (UCC) cannot be simply decomposed in the linear form of its components. The logarithm form of standard (12) is Equation (23). Many studies (e.g., [8]) have



described how to derive a simple empirical equation such as Equation (24) based on the theoretical formula (23). At the same time, our three-asset derivation implies that risk terms should be added to the standard UCC. Therefore, our empirical equation should be (25). The vector $\boldsymbol{X}$ is used to show the explanatory variables of imposed housing rent. Personal disposable income is selected to capture the change in $ln(R)$. Next, in line with previous studies [8], the sum of expected real housing capital return and inflation is approximated by the last nominal house price growth rate ($r_H(t) + \pi(t) = rh(t-1)$). In addition, we estimate the households' relative risk aversion level ($\theta \times HE/CON$), which is 1.015 [29]. Then, we rely on the values of 2010, and $\theta$ is set to be 5.0 ($\approx 1.015 \times CON/HE$).

$$ln(G) = ln(R) - ln(UCC) = ln(R(\boldsymbol{X})) - ln(i - \pi + \delta - r_H) \tag{23}$$

$$lng_t = a_0 + a_1 ln(\boldsymbol{X})_t + a_2 i_t + a_3 rh_{t-1} + \varepsilon_t \tag{24}$$

$$lng_t = \begin{aligned} &a_0 + a_1 ln(\boldsymbol{X})_t + a_2 i_t + a_3 rh_{t-1} \\ &+ a_4 tau1_t + a_5 tau2_t + \varepsilon_t \end{aligned} \tag{25}$$

Since the key variables $lng$ and $lnry$ are non-stationary, co-integration should be noted. The error correction type of Equation (25) has three panel relations: they are: (26)–(28). Heterogeneity is not considered in (26); (27) only considers heterogeneity in the short-term relation; and (28) implies that heterogeneity exists in all of the short-term, long-term, and error-correction processes. In addition, $u_i$ is the individual effect, which is necessary for panel data analysis.

$$\Delta lng_t = \beta \Delta lnry_t + \eta [lng_{t-1} - a_1 lnry_{t-1} - a_2 i_{t-1} \\ - a_3 rh_{t-2} - a_4 tau1_{t-1} - a_5 tau2_{t-1}] + u_i + \varepsilon_t \tag{26}$$

$$\Delta lng_t = \beta_i \Delta lnry_t + \eta_i [lng_{t-1} - a_1 lnry_{t-1} - a_2 i_{t-1} \\ - a_3 rh_{t-2} - a_4 tau1_{t-1} - a_5 tau2_{t-1}] + u_i + \varepsilon_t \tag{27}$$

$$\Delta lng_t = \beta_i \Delta lnry_t + \eta_i [lng_{t-1} - a_{i1} lnry_{t-1} - a_{i2} i_{t-1} \\ - a_{i3} rh_{t-2} - a_{i4} tau1_{t-1} - a_{i5} tau2_{t-1}] + u_i + \varepsilon_t \tag{28}$$

Equations (26)–(28) are regressed by the three regressions in the non-stationary heterogeneous panel model: pooled mean-group (PMG), mean-group (MG), and dynamic fixed-effect (DFE). Due to the allowance of heterogeneity in the regression of PMG and MG, the estimated coefficients are the averages of all panels. Estimated coefficients, as well as their t-values, are given in Table 3.

**Table 3.** Non-stationary Heterogeneous Panel Model, Decomposed UCC, 2002–2018.

| Coefficient | Equation (26), DFE | Equation (27), MG | Equation (28), PMG |
|---|---|---|---|
| $\beta$ (or $E[\beta_i]$) | 0.772 *** (4.4) | 0.785 *** (3.3) | 1.007 *** (5.3) |
| $\eta$ (or $E[\eta_i]$) | −0.383 *** (10.8) | −0.644 *** (10.0) | −0.499 *** (9.7) |
| $a_1$ (or $E[a_{i1}]$) | 0.830 *** (25.7) | 0.906 *** (7.4) | 0.853 *** (55.5) |
| $a_2$ (or $E[a_{i2}]$) | −7.401 *** (3.9) | −4.173 (0.6) | −5.806 *** (6.8) |
| $a_3$ (or $E[a_{i3}]$) | 0.381 *** (3.3) | 0.607 *** (4.0) | 0.168 ** (2.4) |
| $a_4$ (or $E[a_{i4}]$) | −0.367 ** (2.1) | −0.987 * (1.7) | −0.319 *** (3.3) |
| $a_5$ (or $E[a_{i5}]$) | −0.303 ** (2.1) | −0.982 (0.5) | 0.077 (0.9) |
| constant | −0.079 (0.9) | −0.283 * (1.7) | −0.224 *** (5.0) |

Notes: $|t|$ in parentheses; *, **, and *** are significant at the 10%, 5% and 1% levels.

Given the results of Hausman tests among MG&PMG (Prob>chi2 = 0.51), MG&DFE (Prob>chi2 = 1.00) and PMG&DFE (Prob>chi2 = 1.00), Equation (26) is better than (28) and then better than (27), implying that the spatial heterogeneity is not significant, and therefore, all provinces can be regarded as the same in co-integration, including all of the short-term, long-term, and error-correction process.

The UCC is formed based on the estimated values of the DFE shown in Table 3. The ratios among estimated coefficients of the interest rate, expected housing capital return, *tau*1, and *tau*2 ($a_2$, $a_3$, $a_4$, and $a_5$) are approximate to $1 : -0.05 : 0.05 : 0.05$. The annual housing depreciation rate, a necessary component of housing UCC, is commonly assumed to be 0.01 [8]. Thus, standard UCC is organized as (29) for comparison. Our UCC is organized as (30). Since the DFE regression is better than the MG and PMG regressions, the empirical equation with log UCC is (31). The results are given in Table 4.

$$UCC_t = i_t + 0.01 - 0.05 \times rh_{t-1} \tag{29}$$

$$UCC_{3asset,t} = i_t + 0.01 - 0.05 \times rh_{t-1} + 0.05 \times tau1_t + 0.05 \times tau2_t \tag{30}$$

$$\Delta lng_t = \beta \Delta lnry_t + \eta[lng_{t-1} - a_1 lnry_{t-1} - a_2 ln(UCC)_{t-1}] + u_i + \varepsilon_t \tag{31}$$

**Table 4.** Non-stationary Heterogeneous Panel Model, ln(UCC), 2002–2018.

| Coefficient | Equation (31), UCC by (29) | Equation (31), UCC by (30) |
|:---:|:---:|:---:|
| $\beta$ | 0.798 *** (4.5) | 0.815 *** (4.7) |
| $\eta$ | −0.396 *** (11.0) | −0.380 *** (10.7) |
| $a_1$ | 0.812 *** (24.1) | 0.807 *** (23.2) |
| $a_2$ | −0.345 *** (3.6) | −0.410 *** (4.5) |
| constant | −0.583 *** (4.3) | −0.619 *** (5.2) |
| $R^2$ (within) | 0.296 | 0.315 |

Notes: $|t|$ in parentheses; *** is significant at the 1% levels.

The estimated values in Table 4 imply that, when housing market risk premium terms (both *tau*1 and *tau*2) are considered, the significance of $ln(UCC)$ becomes greater ($|t|$ increases from 3.6 to 4.5), and $R^2$ is also increased. This evidence suggests that the three-asset house price derivation is more suitable in house price analysis and forecasting.

In the two-asset model [1], the empirical effect of *tau*2 was discussed, where *tau*2 plays an important role in restricting the price-return positive circle. In contrast, the mean values in Table 1 give us the impression that *tau*1 is significantly smaller than any of the interest rate ($i$), housing capital return ($rh$), and *tau*2. Conditional on their similar marginal effects, the practical effect of *tau*1 seems to be tiny. However, since the correlation between the returns of the two markets practically has a strong cyclical trend rather than being stochastically distributed in the interval $[-1, 1]$, the effect of *tau*1 on real house prices could be apparent.

*3.3. Historical Correlation between Housing and Financial Market Returns*

The correlation between housing and financial market returns ($\rho$) is the key to the housing market risk premium for both $\tau_1$ and $\tau_2$. If the correlation is close to zero, the impact of financial market factors on housing market risk premiums will be tiny, and our $\tau_2$ will decrease to the two-asset derivation; in contrast, if the correlation is close to 1 or $-1$, $\tau_1$ will be significantly non-zero, and $\tau_2$ will converge to zero.

There is a fascinating phenomenon by which, although the correlation between housing and financial market returns is usually close to zero in the long-term measurement, the correlation actually is significantly non-zero in the short-run. Furthermore, the correlation is always negative before financial/economic crises and positive during and after crises. Meen [8] measured the correlation of the UK and showed that, for example, the correlation between house capital return and FTSE return is 0.08 during the period of 1970–2012 but $-0.66$ during the period of 2001–2005 and 0.69 during the period of 2006–2012. Related works [8] have mentioned this phenomenon too. We also measured the correlation between US housing capital return and S&P return and obtained supporting evidence. For China, the long-term correlation was 0.04 during the period of 2000M04–2020M06, and the short-term correlations were calculated based on the sliding window method, with the window width equal to the last 12 months, as shown in Figure 1.

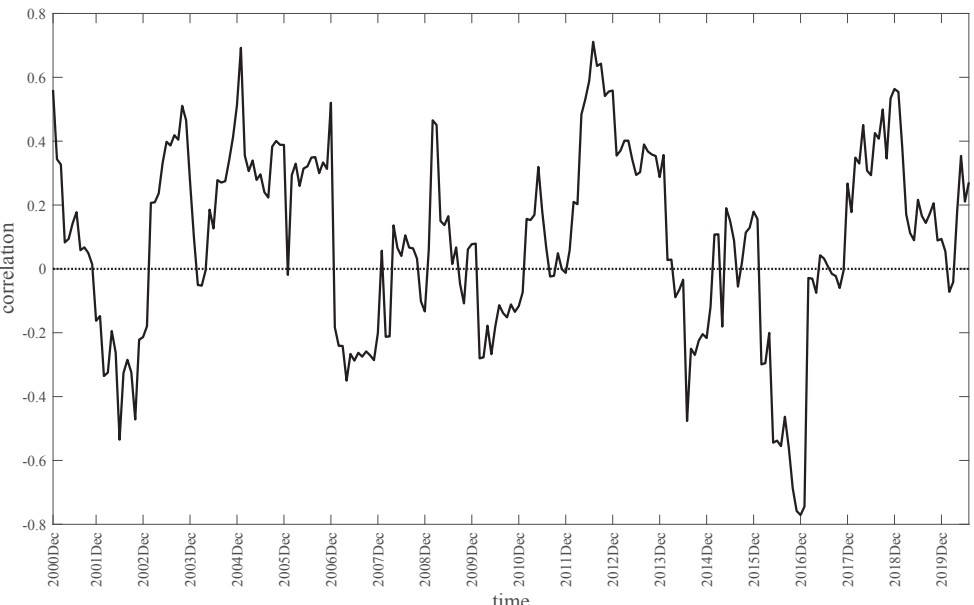

**Figure 1.** Correlation between housing capital return and SSEC return.

Figure 1 indicates that the correlation in the short run can be quite large or quite small ($\approx\pm0.7$). One of the possible interpretations is that, in boom periods, some assets will be more attractive than others, causing changes in households' optimal portfolio decisions ($k^*$) and then causing reversed changes in demand for different assets, but in bust periods, the consumption–investment decision dominates households' behaviours, and therefore, the demands of all risky assets will fall (and recover afterward). If it makes sense that the correlation is negative in the boom period, $\tau_1$ will be negative, and $\tau_2$ will be positive (but small) so that the housing market risk premium could be negative, causing relatively lower UCC and relatively higher house prices; when a crisis occurs, the correlation changes from negative to positive. $\tau_1$ should be positive, and $\tau_2$ should also be positive (but small) so that the housing market risk premium should be positive; therefore, UCC goes up, and house prices go down. In light of this outcome, if the real estate market is the origin of the financial crisis, the changed correlation will aggravate the severity of the financial crisis. In the next section, on the basis of reasonable assumptions about the correlation and other factors, we simulate the influence of the change in correlation on real house prices.

## 4. Simulation Analysis

In this section, we attempt to analyse the dynamic impact of the correlation between housing capital return and financial market return on real house prices under our three-asset model. Generally, housing market risk premiums are generated as the CRRA derivation (19)–(21), housing user cost of capital (UCC) is generated as the empirical Equation (30), and real house prices are generated as the long-term housing price Equation (32), which is the long-run relationship implied by the ECM Equation (31) with the empirical estimations given in Table 4. Note that, since real income will be generated as a non-stationary series, co-integration is in-built when the long-term relationship (32) is applied.

$$lng_t = a_0 + 0.812 \times lnry_t - 0.345 \times ln(UCC)_t \qquad (32)$$

The exogenous shocks are contained by the real income series. The mean and standard deviation of the historical real income growth rate are 0.0776 and 0.0807, respectively. We simply assume that the growth rate of real income is distributed normally and thus generate the exogenous noises, while to simulate the cyclical correlation, the correlation is generated as a sine function ($corr_t = 0.7 \times sin(t \times 2\pi/6.5)$), where $\pi$ is approximate to 3.14, 0.7 is the amplitude, and 6.5 (years) is the cycle period roughly estimated by Figure 1. All of the other details are summarized in Appendix B (Table A1).

Three curves are simulated for the purposes of comparison. The first is the baseline, where exogenous noises are not considered in real income growth. Therefore, house prices increase steadily. Noises are considered in both the second and third. The only difference is that housing market risk premiums are set to be zero in the second, referring to the standard house price model, but the risk premium is considered in the third. It is important to mention that calibration simulation (such as the first curve) of our three-asset model is meaningless because the generation of the risk premium requires volatilities. However, the consideration of noises will lead to a case in which we are not sure whether the change in real house prices is caused by the consideration of the risk premium or the noises. In light of this uncertainty, the stochastic process was replicated 100 times, and we calculated the averages to remove the influence of noise.

Figure 2 shows the key results of the simulation. The first frame shows the stochastic and non-stochastic real income growth rates over 25 years (periods). The second frame shows the generated sine shape correlation. To reflect the properties of risk premium terms, the third frame shows 55 years of results. Doubtlessly, since tau1 is determined by the correlation, it also reflects a sine shape movement. As in the analysis and simulation shown by Wang et al. [1], tau2 is increasing in recent years in China. Tau, the sum of the two, also reflects a strong cyclicity, but compared with tau1, the peaks are wider, and the troughs are narrower, indicating that housing market risk premiums in most of the periods have the relatively strong ability to restrain house price growth; when the financial market and housing market are negatively correlated, the restrictive ability will Decrease and thus cause housing price bubbles. As clearly shown by the fourth frame, when the risk premium terms are involved, there will be obvious periodicity in house prices. Therefore, although housing market risk premiums have a negative impact on real house prices, the natural change is not conducive to stabilizing house price. For instance, real house prices decreased 9.6% in the two years from period 18 to 20 and fell 9.3% in one year from period 25 to 26. These simulated values support our view that the effect of financial market factors on housing market risk premiums, as well as real house prices, cannot be ignored in practice and should be of sufficient concern in both theoretical studies and government policy making.

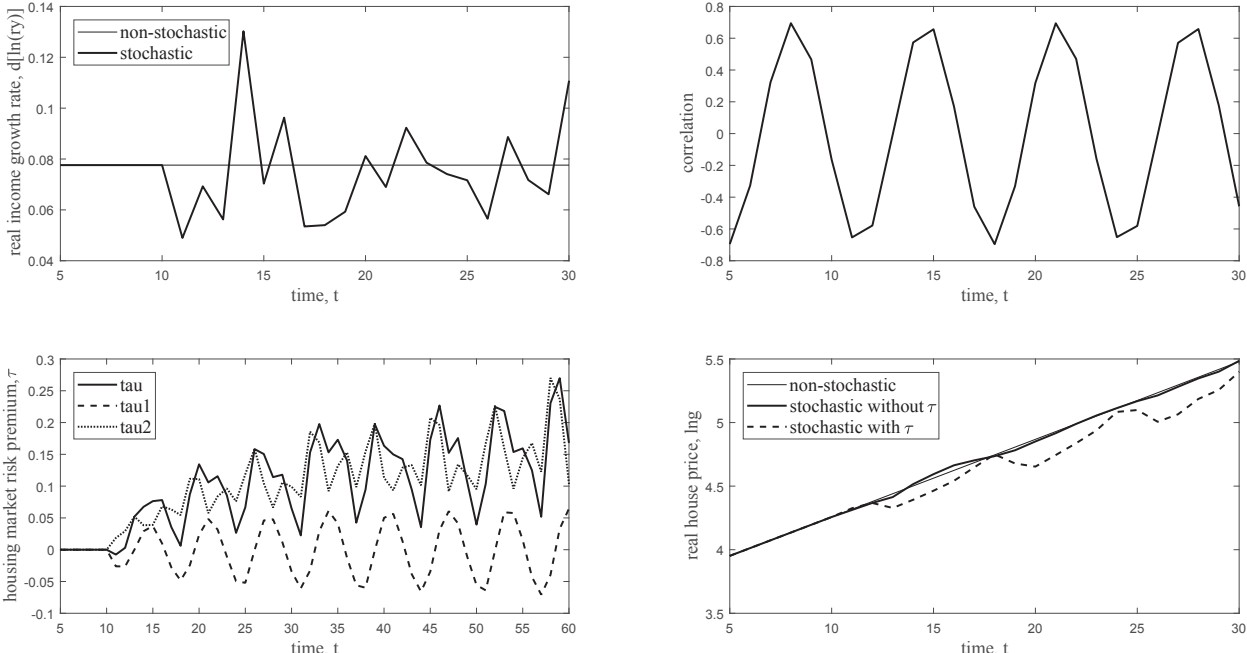

**Figure 2.** Average dynamics under 100 replications (random seed: 1–100).

## 5. Conclusions

This paper aims to analyse the effect of financial market factors on real house prices. By introducing the risky financial asset, we first develop the theoretical framework from a two-asset base to a three-asset model. Theoretical derivations show that three financial market elements—expected return, volatility and the correlation—determine housing market risk premiums and thus affect the housing user cost of capital, as well as real house prices. If correlations between the returns on housing and financial markets are positive, real house prices are negatively affected by expected returns and positively affected by the volatility of the risky financial asset; if the correlation is negative, the effects are reversed. Empirical works have supported the existence of the theoretical derivations of the housing market risk premiums ($\tau_1$ and $\tau_2$), but the estimated marginal effects are relatively small. Statistics have also shown that, despite the correlation between the two markets being close to zero in the long-term measurement, it is significantly negative in boom periods and significantly positive in bust periods. Given reasonable assumptions, our dynamic simulations point out that financial markets cause cyclical changes in real house prices. More specifically, the results show that China's real house prices could decrease by approximately 10% within one to two years in the future if the effects of financial market factors are still as large as they are now.

Therefore, for the purpose of stabilizing housing prices and avoiding housing market collapse, we recommend the following: (i) government should control the money flow from one to another between the housing and financial markets; and (ii) compared with the housing market, financial markets should have an absolute advantage (relatively higher expected return and lower risk).

Although the three-asset model is very complex, it is still incomplete. Housing mortgage debt must necessarily be introduced into the model next. Then, the difference between risk-free interest rates and mortgage interest rates will be allowed. The effect of monetary policy on housing prices can be more precisely analysed.

**Author Contributions:** Conceptualization, Y.W.; methodology, Y.W.; software, Y.W. and J.L.; validation, Y.W. and J.L.; formal analysis, Y.W.; investigation, Y.W.; resources, Y.W.; data curation, Y.W. and J.L.; writing—original draft preparation, Y.W., J.L., Z.Q. and S.S.; writing—review and editing, Y.W., J.L., Z.Q. and S.S.; visualization, Y.W.; supervision, Y.W.; project administration, Y.W.; funding acquisition, Y.W. All authors have read and agreed to the published version of the manuscript.

**Funding:** This research was funded by (China's) Shandong Social Science Planning Research Project (grant number: 19DTJJ02).

**Institutional Review Board Statement:** Not applicable.

**Informed Consent Statement:** Informed consent was obtained from all subjects involved in the study.

**Data Availability Statement:** The data presented in this study are available upon request from the corresponding author.

**Acknowledgments:** This work was supported by the Faculty of Economics and the Centre of Excellence in Econometrics at Chiang Mai University, the China-CASEAN High-Quality Development Research Center in Shandong University of Finance and Economics, and the education plan of the youth and creative talents in Shandong higher education institutions.

**Conflicts of Interest:** The authors declare no conflict of interest.

## Appendix A. Mathematical Derivations

*Appendix A.1. The Expected Utility Three-Asset Housing Life-Cycle Model, Part I*

According to the lifetime utility function (1) and constraints (3)–(6), the households' decisions can be found by maximizing the Hamiltonian function:

$$\mathcal{L}(t) = E\left\{ \int_0^\infty e^{-rt}\left\{ \mu[H(t), C(t)] + \lambda(t)\left[ RY(t) - C(t) \right. \right. \right.$$

$$-g_H(t)\big[\dot{H}(t) + \delta(t)H(t)\big]$$

$$-\big[\dot{A}_f(t) - [i(t) - \pi(t)]A_f(t)\big]$$

$$-g_A(t)\big[\dot{A}_A(t) - [i_d(t) - \pi(t)]A_A(t)\big]\Big]\Big\}dt\bigg\}, \tag{A1}$$

The first-order conditions are

$$\frac{\partial \mathcal{L}(t)}{\partial C(t)} = E\Big\{e^{-rt}\mu_C(t) - e^{-rt}\lambda(t)\Big\} = 0, \tag{A2}$$

$$\frac{\partial \mathcal{L}(t)}{\partial H(t)} = E\Big\{e^{-rt}\mu_H(t) - e^{-rt}\lambda(t)g_H(t)\delta(t) + \frac{d}{dt}[e^{-rt}\lambda(t)g_H(t)]\Big\} = 0, \tag{A3}$$

$$\frac{\partial \mathcal{L}(t)}{\partial A_f(t)} = E\Big\{e^{-rt}\lambda(t)[i(t) - \pi(t)] + \frac{d}{dt}[e^{-rt}\lambda(t)]\Big\} = 0, \tag{A4}$$

$$\frac{\partial \mathcal{L}(t)}{\partial A_A(t)} = E\Big\{e^{-rt}\lambda(t)g_A(t)[i_d(t) - \pi(t)] + \frac{d}{dt}[e^{-rt}\lambda(t)g_A(t)]\Big\} = 0. \tag{A5}$$

Note that all variables of the are certain in time period ($t$) and uncertain in the future ($t + dt$, $t + 2dt$, $t + 3dt$, ...) Therefore, certain elements can be removed to the outside of expectations.

From (A2),

$$\mu_C(t) = \lambda(t). \tag{A6}$$

From (A3),

$$E[e^{-rt}\mu_H(t)] = E[e^{-rt}\lambda(t)g(t)\delta(t)] - E\Big[\frac{d}{dt}[e^{-rt}\lambda(t)g_H(t)]\Big]$$

$$\therefore\ e^{-rt}E[\mu_H(t)] = e^{-rt}E[\lambda(t)]g_H(t)\delta(t) - E\Big[\dot{g}_H(t)e^{-rt}\lambda(t) - g_H(t)\frac{d}{dt}[e^{-rt}\lambda(t)]\Big]$$

$$\therefore\ e^{-rt}E[\mu_H(t)] = e^{-rt}E[\lambda(t)]g_H(t)\delta(t) - e^{-rt}E\Big[\dot{g}_H(t)\lambda(t)\Big] - g_H(t)E\Big[\frac{d}{dt}[e^{-rt}\lambda(t)]\Big]$$

$$\therefore\ E[\mu_H(t)] = g_H(t)\Big\{E[\lambda(t)]\delta(t) - E\Big[r_H(t)\lambda(t)\Big] - e^{rt}E\Big[\frac{d}{dt}[e^{-rt}\lambda(t)]\Big]\Big\} \tag{A7}$$

where $r_H(t) = \dot{g}_H(t)/g_H(t)$, real capital return of housing.

From (A4),

$$[i(t) - \pi(t)]E[\lambda(t)] = -e^{rt}E[\frac{d}{dt}[e^{-rt}\lambda(t)]]. \tag{A8}$$

From (A5),

$$e^{-rt}g_A(t)E\Big[\lambda(t)[i_d(t) - \pi(t)]\Big] = -E\Big[\frac{d}{dt}[e^{-rt}\lambda(t)g_A(t)]\Big]$$

$$\therefore\ e^{-rt}g_A(t)E\Big[\lambda(t)[i_d(t) - \pi(t)]\Big] = -E\Big[\dot{g}_A(t)e^{-rt}\lambda(t) - g_A(t)\frac{d}{dt}[e^{-rt}\lambda(t)]\Big]$$

$$\therefore\ E\Big[[i_d(t) - \pi(t)]\lambda(t)\Big] = -E\Big[r_A(t)\lambda(t)\Big] - e^{rt}E\Big[\frac{d}{dt}[e^{-rt}\lambda(t)]\Big]$$

$$\therefore\ E\Big[[r_A(t) + i_d(t) - \pi(t)]\lambda(t)\Big] = -e^{rt}E\Big[\frac{d}{dt}[e^{-rt}\lambda(t)]\Big] \tag{A9}$$

where $r_A(t) = \dot{g}_A(t)/g_A(t)$ is real capital return of risky financial assets.

Substitute (A6) into (A7)–(A9),

$$\frac{\partial \mathcal{L}(t)}{\partial H(t)}: E[\mu_H(t)] = g_H(t)\Big\{E[\mu_C(t)]\delta(t) - E\Big[r_H(t)\mu_C(t)\Big] - e^{rt}E\Big[\frac{d}{dt}[e^{-rt}\mu_C(t)]\Big]\Big\} \tag{A10}$$

$$\frac{\partial \mathcal{L}(t)}{\partial A_f(t)} : [i(t) - \pi(t)]E[\mu_C(t)] = -e^{rt}E[\frac{d}{dt}[e^{-rt}\mu_C(t)]] \tag{A11}$$

$$\frac{\partial \mathcal{L}(t)}{\partial A_A(t)} : E\Big[[r_A(t) + i_d(t) - \pi(t)]\mu_C(t)\Big] = -e^{rt}E\Big[\frac{d}{dt}[e^{-rt}\mu_C(t)]\Big] \tag{A12}$$

Equations (A10)–(A12) are Equations (7)–(9) in the main text.

*Appendix A.2. The Expected Utility Three-Asset Housing Life-Cycle Model, Part II*

In this part, conditions are Equations (A10)–(A12).

Substitute (A11) into (A12),

$$E\Big[[r_A(t) + i_d(t) - \pi(t)]\mu_C(t)\Big] = [i(t) - \pi(t)]E[\mu_C(t)]$$

$$\frac{E\Big[r_{AD}(t)\mu_C(t)\Big]}{E[\mu_C(t)]} = i(t) - \pi(t) \tag{A13}$$

where $r_{AD}(t) = r_A(t) + i_d(t) - \pi(t)$ and this is Equation (10) in main text.

Or since $E[AB] = E[A]E[B] + Cov[A, B]$,

$$E[r_{AD}(t)] - (i(t) - \pi(t)) = -\frac{Cov\Big[r_{AD}(t), \mu_C(t)\Big]}{E[\mu_C(t)]}.$$

Substitute (A11) into (A10):

$$E[\mu_H(t)] = \Big\{ E[\mu_C(t)]\delta(t) - E[r_H(t)\mu_C(t)] + [i(t) - \pi(t)]E[\mu_C(t)] \Big\}g_H(t)$$

$$\therefore \frac{E[\mu_H(t)]}{E[\mu_C(t)]} = \Big\{ i(t) - \pi(t) + \delta(t) - \frac{E[r_H(t)\mu_C(t)]}{E[\mu_C(t)]} \Big\}g_H(t). \tag{A14}$$

Or since $E[AB] = E[A]E[B] + Cov[A, B]$,

$$\frac{E[\mu_H(t)]}{E[\mu_C(t)]} = \Big\{ i(t) - \pi(t) + \delta(t) - E[r_H(t)] - \frac{Cov[r_H(t), \mu_C(t)]}{E[\mu_C(t)]} \Big\}g_H(t). \tag{A15}$$

Equations (A14) and (A15) are Equations (11)–(12) in the main text.

*Appendix A.3. The Three-Asset Expected Utility Housing Life-Cycle Model under CARA Utility*

In this part, conditions are Equations (13) and (14).

Since $r_H$ and $r_{AD}$ are joint normally distributed, the probability density function of the standard joint normal distribution between $z_1$ and $z_2$ is used, given as:

$$\phi(z_1, z_2) = \frac{1}{2\pi\sqrt{1-\rho^2}}exp\Big\{ -\frac{1}{2(1-\rho^2)}[z_1^2 - 2\rho z_1 z_2 + z_2^2] \Big\} \tag{A16}$$

where $\rho$ is the correlation between $z_1$ and $z_2$.

Equations (3)–(6) can be organized as one equation:

$$C(t) = RY(t) + i(t)A_f(t) + i_d(t)A_A(t) - g_H(t)[\dot{H}(t) + \delta(t)H(t)]$$

$$-[\dot{A}(t) + \pi(t)A(t)] - g_A(t)[\dot{A}_A(t) + \pi(t)A_A(t)]$$

or

$$C(t) \approx RY(t) + [r_H(t) - \delta(t)]g_H(t)H(t)$$

$$+[i(t) - \pi(t)]A_f(t) + r_{AD}(t)g_A(t)A_A(t) \tag{A17}$$

This equation implies that, if the quantities of households' housing and non-housing assets are unchanged at the moment, all of their labour income, housing net yield and non-housing financial yield will be consumed. Further, since the housing capital return ($r_H(t)$) and the total return of risky financial assets ($r_{AD}(t)$) are uncertain, households' consumption (A17) can be divided into certain and uncertain parts as:

$$C(t) = RY(t) + [r_H^e(t) - \delta(t)]g_H(t)H(t) + g_H(t)H(t)\sigma_H(t)z_1$$

$$+ [i(t) - \pi(t)]A_f(t) + r_{AD}^e(t)g_A(t)A_A(t) + g_A(t)A_A(t)\sigma_{AD}(t)z_2$$

Therefore,

$$E[C(t)] \equiv C^e(t) = RY(t) + [r_H^e(t) - \delta(t)]g_H(t)H(t)$$

$$+ [i(t) - \pi(t)]A_f(t) + r_{AD}^e(t)g_A(t)A_A(t)$$

or

$$C(t) = C^e(t) + g_H(t)H(t)\sigma_H(t)z_1 + g_A(t)A_A(t)\sigma_{AD}(t)z_2 \tag{A18}$$

Remember that, to derive a specific solution for Equation (12) requires solving $E[\mu_C(t)]$, $E[r_H(t)\mu_C(t)]$ and $E[r_{AD}(t)\mu_C(t)]$.

　　Step 1: The derivation of $E[\mu_C(t)]$.
　　Under the specific CARA utility function (14),

$$E[\mu_C(t)] = E[\varphi exp[-\varphi C(t)]].$$

Substitute (A18),

$$E[\mu_C(t)] = E[\varphi exp[-\varphi C^e(t) - \varphi g_H(t)H(t)\sigma_H(t)z_1 - \varphi g_A(t)A_A(t)\sigma_{AD}(t)z_2]]$$

$$\therefore\ E[\mu_C(t)] = \varphi exp[-\varphi C^e(t)] \times E[exp[-\varphi g_H(t)H(t)\sigma_H(t)z_1 - \varphi g_A(t)A_A(t)\sigma_{AD}(t)z_2]]$$

Based on the standard joint normal distribution,

$$E[\mu_C(t)] = \varphi exp[-\varphi C^e(t)] \times \iint_{-\infty}^{+\infty} \Big\{ exp[-\varphi g_H(t)H(t)\sigma_H(t)z_1$$

$$- \varphi g_A(t)A_A(t)\sigma_{AD}(t)z_2]\Big\}\phi(z_1,z_2)dz_1dz_2$$

Substituting the probability density function (A16),

$$E[\mu_C(t)] = \varphi exp[-\varphi C^e(t)] \times \iint_{-\infty}^{+\infty} \Big\{ exp[-\varphi g_H(t)H(t)\sigma_H(t)z_1 - \varphi g_A(t)A_A(t)$$

$$\sigma_{AD}(t)z_2]\Big\} \frac{1}{2\pi\sqrt{1-\rho^2(t)}} exp\Big\{ -\frac{1}{2(1-\rho^2(t))}[z_1^2 - 2\rho(t)z_1z_2 + z_2^2]\Big\}dz_1dz_2$$

$$\therefore\ E[\mu_C(t)] = \varphi exp[-\varphi C^e(t)] \times \iint_{-\infty}^{+\infty} \frac{1}{2\pi\sqrt{1-\rho^2(t)}} exp\Big\{ -\frac{1}{2(1-\rho^2(t))}[z_1^2 - 2\rho(t)z_1z_2$$

$$+ z_2^2 + 2(1-\rho^2(t))\varphi g_H(t)H(t)\sigma_H(t)z_1 + 2(1-\rho^2(t))\varphi g_A(t)A_A(t)\sigma_{AD}(t)z_2]\Big\}dz_1dz_2$$

Because of the mathematical law that

$$x^2 - 2\rho xy + y^2 + 2(a - \rho b)x + 2(b - \rho a)y + (a^2 - 2\rho ab + b^2)$$

$$= (x+a)^2 - 2\rho(x+a)(y+b) + (y+b)^2.$$

Let, $x = z_1$; $a - \rho(t)b = (1 - \rho^2(t))\varphi g_H(t)H(t)\sigma_H(t)$; $y = z_2$;
$b - \rho(t)a = (1 - \rho^2(t))\varphi g_A(t)A_A(t)\sigma_{AD}(t)$.

Therefore,
$$
\begin{cases}
a &= \varphi\Big[g_H(t)H(t)\sigma_H(t) + \rho(t)g_A(t)A_A(t)\sigma_{AD}(t)\Big] \\
b &= \varphi\Big[\rho(t)g_H(t)H(t)\sigma_H(t) + g_A(t)A_A(t)\sigma_{AD}(t)\Big]
\end{cases}.
$$

Thus, $a^2 - 2\rho ab + b^2 = \varphi^2(1 - \rho^2(t))M(t)$

where $M(t) = \big[g_H(t)H(t)\sigma_H(t)\big]^2 + 2\rho(t)\big[g_H(t)H(t)\sigma_H(t)\big]\big[g_A(t)A_A(t)\sigma_{AD}(t)\big] + \big[g_A(t)A_A(t)\sigma_{AD}(t)\big]^2.$

Therefore, $E[\mu_C(t)] = \varphi exp[-\varphi C^e(t)] \times \iint_{-\infty}^{+\infty} \frac{1}{2\pi\sqrt{1-\rho^2(t)}} exp\Big\{ - \frac{1}{2(1-\rho^2(t))}$

$\Big[(x+a)^2 - 2\rho(x+a)(y+b) + (y+b)^2 - (a^2 - 2\rho ab + b^2)\Big]\Big\}dz_1 dz_2$

$\therefore\ E[\mu_C(t)] = \varphi exp[-\varphi C^e(t)] \times exp\Big\{\frac{a^2 - 2\rho ab + b^2}{2(1-\rho^2(t))}\Big\} \iint_{-\infty}^{+\infty} \frac{1}{2\pi\sqrt{1-\rho^2(t)}}$

$exp\Big\{ - \frac{1}{2(1-\rho^2(t))}\Big[(x+a)^2 - 2\rho(x+a)(y+b) + (y+b)^2\Big]\Big\}dz_1 dz_2$

$\therefore\ E[\mu_C(t)] = \varphi exp[-\varphi C^e(t)] \times exp\Big\{\frac{\varphi^2(1-\rho^2(t))M(t)}{2(1-\rho^2(t))}\Big\}$

$\times \iint_{-\infty}^{+\infty} \phi[(z_1 + a), (z_2 + b)]dz_1 dz_2$

$$
\therefore\ E[\mu_C(t)] = \varphi exp[-\varphi C^e(t)] \times exp\Big\{\tfrac{1}{2}\varphi^2 M(t)\Big\}. \tag{A19}
$$

Step 2: The derivation of $E[r_H(t)\mu_C(t)]$.

$E[r_H(t)\mu_C(t)]$

$= E\Big[[r_H^e(t) + \sigma_H(t)z_1]\mu_C(t)\Big]$

$= r_H^e(t)E[\mu_C(t)] + E\Big\{\sigma_H(t)z_1\mu_C(t)\Big\}$

$= r_H^e(t)E[\mu_C(t)] + E\Big\{\sigma_H(t)z_1\varphi exp\Big[ - \varphi C(t)\Big]\Big\}$

$= r_H^e(t)E[\mu_C(t)] + E\Big\{\sigma_H(t)z_1\varphi exp\Big[ - \varphi\big[C^e(t) + g_H(t)H(t)\sigma_H(t)z_1$

$+g_A(t)A_A(t)\sigma_{AD}(t)z_2\big]\Big]\Big\}$

$= r_H^e(t)E[\mu_C(t)] + \iint_{-\infty}^{+\infty}\Big\{\sigma_H(t)z_1\varphi exp\Big[ - \varphi\big[C^e(t) + g_H(t)H(t)\sigma_H(t)z_1$

$+g_A(t)A_A(t)\sigma_{AD}(t)z_2\big]\Big]\Big\}\phi(z_1, z_2)dz_1 dz_2$

$= r_H^e(t)E[\mu_C(t)] + \iint_{-\infty}^{+\infty}\Big\{\sigma_H(t)z_1\varphi exp\Big[ - \varphi C^e(t) - \varphi g_H(t)H(t)\sigma_H(t)z_1$

$-\varphi g_A(t)A_A(t)\sigma_{AD}(t)z_2\Big]\Big\}\frac{1}{2\pi\sqrt{1-\rho^2(t)}}exp\Big\{ - \frac{1}{2(1-\rho^2(t))}[z_1^2 - 2\rho z_1 z_2 + z_2^2]\Big\}dz_1 dz_2$

$= r_H^e(t)E[\mu_C(t)] + \iint_{-\infty}^{+\infty}\Big\{\sigma_H(t)z_1\varphi \times exp[-\varphi C^e(t)]\frac{1}{2\pi\sqrt{1-\rho^2(t)}}exp\Big[ - \varphi g_H(t)H(t)\sigma_H(t)z_1$

$-\varphi g_A(t)A_A(t)\sigma_{AD}(t)z_2\Big]\Big\}exp\Big\{ - \frac{1}{2(1-\rho^2(t))}[z_1^2 - 2\rho(t)z_1 z_2 + z_2^2]\Big\}dz_1 dz_2$

$= r_H^e(t)E[\mu_C(t)] + \sigma_H(t)\varphi \times exp[-\varphi C^e(t)]\iint_{-\infty}^{+\infty}\Big\{\frac{1}{2\pi\sqrt{1-\rho^2(t)}}z_1 \times exp\Big\{ - \frac{1}{2(1-\rho^2(t))}$

$\Big[z_1^2 - 2\rho(t)z_1 z_2 + z_2^2 + 2(1-\rho^2(t))\varphi g_H(t)H(t)\sigma_H(t)z_1 + 2(1-\rho^2(t))\varphi g_A(t)A_A(t)\sigma_{AD}$

$(t)z_2\Big]\Big\}dz_1 dz_2.$

Because of the mathematical law that $\iint \phi(x,y)dxdy = \int\Big[\int \phi(x,y)dy\Big]dx$, where $\int \phi(x,y)dy = \phi_x(x) = \frac{1}{\sqrt{2\pi}}exp\Big\{ - \frac{1}{2}x^2\Big\}$ is the marginal probability of $x$.

(Kenney and Keeping, 1951, p. 202).

Then,

$E[r_H(t)\mu_C(t)] = r_H^e(t)E[\mu_C(t)] + \varphi exp[-\varphi C^e(t)]\sigma_H(t) \times exp\Big\{\tfrac{1}{2}\varphi^2 M(t)\Big\}$

$\int_{-\infty}^{+\infty} z_1 f_{z_1}(z_1)dz_1$, where $f_{z_1}(z_1) = \frac{1}{\sqrt{2\pi}}exp\Big\{\tfrac{1}{2}\big[z_1 + \varphi g_H(t)H(t)\sigma_H(t) + \rho(t)\varphi g_A(t)A_A(t)\sigma_{AD}(t)\big]^2\Big\}$.

Therefore,

$$E[r_H(t)\mu_C(t)] = r_H^e(t)E[\mu_C(t)] + \varphi exp[-\varphi C^e(t)]\sigma_H(t) \times exp\left\{\tfrac{1}{2}\varphi^2 M(t)\right\}$$
$$\times \big[ -\varphi g_H(t)H(t)\sigma_H(t) - \rho(t)\varphi g_A(t)A_A(t)\sigma_{AD}(t)\big]$$

$$\therefore\ E[r_H(t)\mu_C(t)] = r_H^e(t)E[\mu_C(t)] \\ + \big[ -\varphi g_H(t)H(t)\sigma_H^2(t) - \varphi g_A(t)A_A(t)[\rho(t)\sigma_H(t)\sigma_{AD}(t)]\big]E[\mu_C(t)]. \tag{A20}$$

Step 3: The derivation of $E[r_{AD}(t)\mu_C(t)]$.

Similarly,

$$E[r_{AD}(t)\mu_C(t)] = r_{AD}^e(t)E[\mu_C(t)] + \varphi exp[-\varphi C^e(t)]\sigma_{AD}(t) \times exp\left\{\tfrac{1}{2}\varphi^2 M(t)\right\}$$
$$\times \big[ -\rho(t)\varphi g_H(t)H(t)\sigma_H(t) - \varphi g_A(t)A_A(t)\sigma_{AD}(t)\big]$$

$$\therefore\ E[r_{AD}(t)\mu_C(t)] = r_{AD}^e(t)E[\mu_C(t)] \\ + \big[ -\varphi g_A(t)A_A(t)\sigma_{AD}^2(t) - \varphi g_H(t)H(t)[\rho(t)\sigma_H(t)\sigma_{AD}(t)]\big]E[\mu_C(t)]. \tag{A21}$$

Step 4: Calculate the UCC as well as tau ($\tau_{CARA}$) under CARA.

Substitute (A21) into (10):

$$i(t) - \pi(t) = r_{AD}^e(t) + \big[ -\varphi g_A(t)A_A(t)\sigma_{AD}^2(t) - \varphi g_H(t)H(t)[\rho(t)\sigma_H(t)\sigma_{AD}(t)]\big]$$

$$\therefore\ \varphi g_A(t)A_A(t) = \frac{r_{AD}^e(t) - (i(t) - \pi(t))}{\sigma_{AD}^2(t)} - \varphi g_H(t)H(t)\left[\rho(t)\frac{\sigma_H(t)}{\sigma_{AD}(t)}\right]. \tag{A22}$$

Substitute (A20) into (11):

$$\frac{E[\mu_H(t)]}{E[\mu_C(t)]} = \left\{ i(t) - \pi(t) + \delta(t) - r_H^e(t) + \tau_{CARA}(t)\right\}g_H(t)$$

where $\tau_{CARA}(t) = \varphi g_H(t)H(t)\sigma_H^2(t) + \varphi g_A(t)A_A(t)[\rho(t)\sigma_H(t)\sigma_{AD}(t)]$.

Substitute (A22) into $\tau_{CARA}(t)$; then we have:

$$\tau_{CARA}(t) = \tau_{1,CARA}(t) + \tau_{2,CARA}(t) \tag{A23}$$

$$\tau_{1,CARA}(t) = [r_{AD}(t) - (i(t) - \pi(t))] \times [\rho(t)\sigma_H(t)/\sigma_{AD}(t)] \tag{A24}$$

$$\tau_{2,CARA}(t) = \varphi g_H(t)H(t)(1 - \rho^2(t))\sigma_H^2(t) \tag{A25}$$

Equations (A23)–(A25) are Equations (16)–(18) in the main text.

*Appendix A.4. The Three-Asset Expected Utility Housing Life-Cycle Model under CRRA Utility*

In this part, conditions are Equations (13), (15), (A17) and (A18).

Similarly, to derive the specific solution of Equation (12) requires solving $E[\mu_C(t)]$, $E[r_H(t)\mu_C(t)]$ and $E[r_{AD}(t)\mu_C(t)]$.

Step 1: The derivation of $E[\mu_C(t)]$.
$E[\mu_C(t)] = \iint_{-\infty}^{+\infty} \mu_C(t)\phi(z_1, z_2)dz_1dz_2$
$\therefore\ E[\mu_C(t)] = \iint_{-\infty}^{+\infty} C(t)^{-\theta}\phi(z_1, z_2)dz_1dz_2$
where $\phi(z_1, z_2)$ is defined in Appendix A.3.

Let us take a second-order Taylor expansion for $C(t)^{-\theta}$, where $C(t) = C^e(t)$:
$C(t)^{-\theta} \approx C^e(t)^{-\theta} - \theta C^e(t)^{-\theta-1}[C(t) - C^e(t)]$
$+ \tfrac{1}{2}\theta(\theta + 1)C^e(t)^{-\theta-2}[C(t) - C^e(t)]^2$
Then,
$E[\mu_C(t)] \approx \iint_{-\infty}^{+\infty} \left\{ C^e(t)^{-\theta} - \theta C^e(t)^{-\theta-1}[C(t) - C^e(t)] \right.$
$+ \tfrac{1}{2}\theta(\theta + 1)C^e(t)^{-\theta-2}[C(t) - C^e(t)]^2 \Big\}\phi(z_1, z_2)dz_1dz_2$
$\therefore\ E[\mu_C(t)] \approx \iint_{-\infty}^{+\infty} \left\{ C^e(t)^{-\theta} - \theta C^e(t)^{-\theta-1}[g_H(t)H(t)\sigma_H(t)z_1 + g_A(t)A_A(t)\sigma_{AD}(t)z_2] \right.$
$+ \tfrac{1}{2}\theta(\theta + 1)C^e(t)^{-\theta-2}[g_H(t)H(t)\sigma_H(t)z_1 + g_A(t)A_A(t)\sigma_{AD}(t)z_2]^2 \Big\}\phi(z_1, z_2)dz_1dz_2$ Since the mathematical laws

$\iint_{-\infty}^{+\infty} c\phi(x,y)dxdy = c$ (constant);

$\iint_{-\infty}^{+\infty} x\phi(x,y)dxdy = 0$ (mean of the joint standard normal distribution);

$\iint_{-\infty}^{+\infty} x^2\phi(x,y)dxdy = 1$ (variance of the joint standard normal distribution); and

$\iint_{-\infty}^{+\infty} xy\phi(x,y)dxdy = \rho$ (covariance of the joint standard normal distribution);

Then,

$$
\begin{aligned}
E[\mu_C(t)] &\approx C^e(t)^{-\theta} + \tfrac{1}{2}\theta(\theta+1)C^e(t)^{-\theta-2}\{[g_H(t)H(t)\sigma_H(t)]^2 \\
&+[g_A(t)A_A(t)\sigma_{AD}(t)]^2 + 2\rho(t)[g_H(t)H(t)\sigma_H(t)][g_A(t)A_A(t)\sigma_{AD}(t)]\}.
\end{aligned}
\tag{A26}
$$

Step 2: The derivation of $E[r_H(t)\mu_C(t)]$.

$E[r_H(t)\mu_C(t)] = \iint_{-\infty}^{+\infty} r_H(t)\mu_C(t)\phi(z_1, z_2)dz_1 dz_2$

$\therefore\ E[r_H(t)\mu_C(t)] = \iint_{-\infty}^{+\infty} r_H(t)C(t)^{-\theta}\phi(z_1, z_2)dz_1 dz_2$

Let us take a second-order Taylor expansion for $r_H(t)C(t)^{-\theta}$, where $C(t) = C^e(t)$ and $r_H(t) = r_H^e(t)$:

$$
\begin{aligned}
r_H(t)C(t)^{-\theta} &\approx r_H^e(t)C^e(t)^{-\theta} \\
&+r_H^e(t)(-\theta)C^e(t)^{-\theta-1}[C(t) - C^e(t)] \\
&+C^e(t)^{-\theta}[r_H(t) - r_H^e(t)] \\
&+\tfrac{1}{2}\theta(\theta+1)r_H^e(t)C^e(t)^{-\theta-2}[C(t) - C^e(t)]^2 \\
&+0 \times [r_H(t) - r_H^e(t)]^2 \\
&+(-\theta)C^e(t)^{-\theta-1}[C(t) - C^e(t)][r_H(t) - r_H^e(t)]
\end{aligned}
$$

$$
\begin{aligned}
r_H(t)C(t)^{-\theta} &\approx r_H^e(t)C^e(t)^{-\theta} \\
&+r_H^e(t)(-\theta)C^e(t)^{-\theta-1}[g_H(t)H(t)\sigma_H(t)z_1 + g_A(t)A_A(t)\sigma_{AD}(t)z_2] \\
&+C^e(t)^{-\theta}[\sigma_H(t)z_1] \\
&+\tfrac{1}{2}\theta(\theta+1)r_H^e(t)C^e(t)^{-\theta-2}[g_H(t)H(t)\sigma_H(t)z_1 + g_A(t)A_A(t)\sigma_{AD}(t)z_2]^2 \\
&+(-\theta)C^e(t)^{-\theta-1}[g_H(t)H(t)\sigma_H(t)z_1 + g_A(t)A_A(t)\sigma_{AD}(t)z_2][\sigma_H(t)z_1]
\end{aligned}
$$

Therefore,

$$
\begin{aligned}
E[r_H(t)\mu_C(t)] &\approx r_H^e(t)C^e(t)^{-\theta} \\
&+\tfrac{1}{2}\theta(\theta+1)r_H^e(t)C^e(t)^{-\theta-2}\{[g_H(t)H(t)\sigma_H(t)]^2 + [g_A(t)A_A(t)\sigma_{AD}(t)]^2 \\
&+2\rho(t)[g_H(t)H(t)\sigma_H(t)][g_A(t)A_A(t)\sigma_{AD}(t)]\} \\
&+(-\theta)C^e(t)^{-\theta-1}[g_H(t)H(t)\sigma_H^2(t) + \rho(t)g_A(t)A_A(t)\sigma_H(t)\sigma_{AD}(t)]
\end{aligned}
\tag{A27}
$$

Step 3: The derivation of $E[r_{AD}(t)\mu_C(t)]$.

Similarly,

$$
\begin{aligned}
E[r_{AD}(t)\mu_C(t)] &\approx r_{AD}^e(t)C^e(t)^{-\theta} \\
+\tfrac{1}{2}\theta(\theta+1)r_{AD}^e(t)C^e(t)^{-\theta-2}\{[g_H(t)H(t)\sigma_H(t)]^2 &+ [g_A(t)A_A(t)\sigma_{AD}(t)]^2 \\
+2\rho(t)[g_H(t)H(t)\sigma_H(t)][g_A(t)A_A(t)\sigma_{AD}(t)]\} & \\
+(-\theta)C^e(t)^{-\theta-1}[g_A(t)A_A(t)\sigma_{AD}^2(t) &+ \rho(t)g_H(t)H(t)\sigma_H(t)\sigma_{AD}(t)]
\end{aligned}
\tag{A28}
$$

Step 4: Calculate the UCC, as well as tau ($\tau_{CRRA}$) under CRRA.

Substitute (A26) into (A28):

$$
\begin{aligned}
E[r_{AD}(t)\mu_C(t)] &\approx r_{AD}^e(t)E[\mu_C(t)] - \theta C^e(t)^{-\theta-1}[g_A(t)A_A(t)\sigma_{AD}^2(t) \\
&+g_H(t)H(t)\rho(t)\sigma_H(t)\sigma_{AD}(t)]
\end{aligned}
\tag{A29}
$$

Substitute (A29) into (10):

$$
\frac{[r_{AD}^e(t) - (i(t) - \pi(t))]E[\mu_C(t)]}{\theta C^e(t)^{-\theta-1}} \approx g_A(t)A_A(t)\sigma_{AD}^2(t) + g_H(t)H(t)\rho(t)\sigma_H(t)\sigma_{AD}(t)
\tag{A30}
$$

Substitute (A26) into (A27):

$$E[r_H(t)\mu_C(t)] \approx r_H^e(t)E[\mu_C(t)] - \theta C^e(t)^{-\theta-1}[g_H(t)H(t)\sigma_H^2(t)$$

$$+\rho(t)g_A(t)A_A(t)\sigma_H(t)\sigma_{AD}(t)] \tag{A31}$$

Substitute (A30) into (A31):

$$\frac{E[r_H(t)\mu_C(t)]}{E[\mu_C(t)]} \approx r_H^e(t) - [r_{AD}(t) - (i(t) - \pi(t))] \times [\rho(t)\sigma_H(t)/\sigma_{AD}(t)]$$

$$-\frac{\theta C^e(t)^{-\theta-1}}{E[\mu_C(t)]}g_H(t)H(t)(1 - \rho^2(t))\sigma_H^2(t) \tag{A32}$$

If expectations are ignored, $\{\theta C^e(t)^{-\theta-1}\}/\{E[C(t)^{-\theta}]\} = \theta/C(t)$, it will be the Pratt-Arrow coefficient for CRRA utility under certainty ($-\mu_C''(t)/\mu_C'(t) = \theta/C(t)$). Under this approximation, the housing market risk ($\tau_{CRRA}$) is much more easily understood. Note that the coefficient of the uncertain utility ($\{\theta C^e(t)^{-\theta-1}\}/\{E[C(t)^{-\theta}]\}$) is still solvable, but since (i) national risk attitude is relatively stable, (ii) its difference from $\theta/C(t)$ is tiny, and (iii) it has a pattern similar to that of $\tau_{CARA}$ derivation, the approximation is more meaningful. Under the approximation, substitute (A32) into (11):

$$\frac{E[\mu_H(t)]}{E[\mu_C(t)]} \approx \left\{i(t) - \pi(t) + \delta(t) - r_H^e(t) + \tau_{CRRA}(t)\right\}g_H(t)$$

where

$$\tau_{CRRA}(t) = \tau_{1,CRRA}(t) + \tau_{2,CRRA}(t) \tag{A33}$$

$$\tau_{1,CRRA}(t) = [r_{AD}(t) - (i(t) - \pi(t))] \times [\rho(t)\sigma_H(t)/\sigma_{AD}(t)] \tag{A34}$$

$$\tau_{2,CRRA}(t) \approx \frac{\theta}{C^e(t)}g_H(t)H(t)(1 - \rho^2(t))\sigma_H^2(t) \tag{A35}$$

Equations (A33)–(A35) are Equations (19)–(21) in the main text.

*Appendix A.5. Derivation of the Optimal Ratio of Risk Financial Assets to Housing Assets from Our Three-Asset Model*

$$\frac{E[\mu_H(t)]}{E[\mu_C(t)]} = MRS_{H,C}(t) = R(t) \tag{A36}$$

Equation (A36) is the arbitrage condition. The marginal rate of substitution (MRS) between housing and the consumption good is equal to the imputed rental price of housing services ($R$), indicating that households' utility is unchanged if their housing services are cut by one unit, but they obtain some non-housing goods as compensation (more details are provided in Wang et al., 2020).

Since Equation (11), the expected total housing return will be

$$r_{HT}^e(t) = r_H^e(t) + R(t)/g(t) - \delta(t) = i(t) - \pi(t) - \frac{E[r_H(t)\mu_C(t)]}{E[\mu_C(t)]} + r_H^e(t) \tag{A37}$$

Part 1: The three-asset model under the CARA utility:
The two conditions are (A20) and (A21). Substitute (A20) into (A37):

$$\frac{r_{HT}^e(t) - [i(t) - \pi(t)]}{\sigma_H(t)} = \varphi g_H(t)H(t)\sigma_H(t) + \varphi g_A(t)A_A(t)[\rho(t)\sigma_{AD}(t)] \tag{A38}$$

Substitute (A21) into (10):

$$\frac{r^e_{AD}(t) - [i(t) - \pi(t)]}{\sigma_{AD}(t)} = \varphi g_A(t) A_A(t) \sigma_{AD}(t) + \varphi g_H(t) H(t)[\rho(t)\sigma_H(t)] \tag{A39}$$

Now, let $\varphi g_H(t)H(t)$ and $\varphi g_A(t)A_A(t)$ be the two unknowns of the binary Equations (A38) and (A39). Then,

$$k^*(t) = \frac{g_A(t)A_A(t)}{g_H(t)H(t)} = \frac{[r^e_{AD}(t) - [i(t) - \pi(t)]]\sigma^2_H(t) + [r^e_{HT}(t) - [i(t) - \pi(t)]]\sigma_{AH}(t)}{[r^e_{HT}(t) - [i(t) - \pi(t)]]\sigma^2_{AD}(t) + [r^e_{AD}(t) - [i(t) - \pi(t)]]\sigma_{AH}(t)} \tag{A40}$$

where $\sigma_{AH}(t) = \rho(t)\sigma_H(t)\sigma_{AD}(t)$ is the covariance of the two returns.

Part 2: The three-asset model under the CRRA utility:
The two conditions are (A29) and (A31). Substitute (A29) into (A37):

$$\frac{r^e_{HT}(t) - [i(t) - \pi(t)]}{\sigma_H(t)} = \frac{\theta C^e(t)^{-\theta-1}}{E[\mu_C(t)]}[g_H(t)H(t)\sigma_H(t) + g_A(t)A_A(t)\rho(t)\sigma_{AD}(t)] \tag{A41}$$

Substitute (A31) into (10):

$$\frac{r^e_{AD}(t) - [i(t) - \pi(t)]}{\sigma_{AD}(t)} = \frac{\theta C^e(t)^{-\theta-1}}{E[\mu_C(t)]}[g_H(t)H(t)\rho(t)\sigma_H(t) + g_A(t)A_A(t)\sigma_{AD}(t)] \tag{A42}$$

Similarly, let $\frac{\theta C^e(t)^{-\theta-1}}{E[\mu_C(t)]}g_H(t)H(t)$ and $\frac{\theta C^e(t)^{-\theta-1}}{E[\mu_C(t)]}g_A(t)A_A(t)$ be the two unknowns of the binary Equations (A41) and (A42). Then,

$$k^*(t) = \frac{g_A(t)A_A(t)}{g_H(t)H(t)} = \frac{[r^e_{AD}(t) - [i(t) - \pi(t)]]\sigma^2_H(t) + [r^e_{HT}(t) - [i(t) - \pi(t)]]\sigma_{AH}(t)}{[r^e_{HT}(t) - [i(t) - \pi(t)]]\sigma^2_{AD}(t) + [r^e_{AD}(t) - [i(t) - \pi(t)]]\sigma_{AH}(t)} \tag{A43}$$

where $\sigma_{AH}(t) = \rho(t)\sigma_H(t)\sigma_{AD}(t)$ is the covariance of the two returns.
Equations (A40) and (A43) are the same and they are Equation (22) in the main text.

*Appendix A.6. Derivation of the Optimal Ratio of Risky Financial Assets to Housing Assets from the Mean-Variance Model*

The mean-variance model is the predecessor of the standard CAPM, and the risk-return relationship is clearly described by the mean-variance model when all assets are categorized into three kinds: two are risky assets, and one is risk-free assets. Simply assume that there are two risky assets A and B, and one risk-free asset, RF, in the market. The means and variances (expected returns and risks) of these three assets are:

$$
\begin{array}{lll}
\text{A:} & E[r_A] = r^e_A & \text{and } Var[r_A] = \sigma^2_A \\
\text{B:} & E[r_B] = r^e_B & \text{and } Var[r_B] = \sigma^2_B \\
\text{RF:} & E[r_{RF}] = r_{RF} & \text{and } Var[r_{RF}] = 0
\end{array}
$$

Investors will invest in these three assets; thus, the shares of money invested in A, B and RF are $d_1$, $d_2$ and $d_3$. respectively The mean and variance of the portfolio are:

$$r^e_T = d_1 r^e_A + d_2 r^e_B + d_3 r_{RF} \tag{A44}$$

$$\sigma^2_T = d^2_1 \sigma^2_A + d^2_2 \sigma^2_B + 2d_1 d_2 \sigma_{AB} \tag{A45}$$

(where $d_1 + d_2 + d_3 = 1$; $\sigma_{AB}$ is the covariance of $r_A$ and $r_B$).

Given a variety of combinations ($d_i$), investors' portfolio decisions will be more efficient when the expected total return ($r^e_T$) is the maximum from all plots that have the same total risk ($\sigma^2_T$). Thus, maximize (A44) subject to (A45), and $d_1 + d_2 + d_3 = 1$.

Lagrangian:

$$\mathcal{L} = d_1 r_A^e + d_2 r_B^e + (1 - d_1 - d_2) r_{RF} + \lambda [d_1^2 \sigma_A^2 + d_2^2 \sigma_B^2 + 2 d_1 d_2 \sigma_{AB} - \sigma_T^2]$$

First order conditions:

$$\frac{\partial \mathcal{L}}{\partial d_1} = r_A^e - r_{RF} + 2\lambda d_1 \sigma_A^2 + 2\lambda d_2 \sigma AB = 0 \qquad (A46)$$

$$\frac{\partial \mathcal{L}}{\partial d_2} = r_B^e - r_{RF} + 2\lambda d_2 \sigma_A^2 + 2\lambda d_1 \sigma AB = 0 \qquad (A47)$$

Combine (A46) and (A47):

$$d_1 = \frac{-[r_A^e - r_{RF}]\sigma_B^2 + [r_B^e - r_{RF}]\sigma_{AB}}{2\lambda(\sigma_A^2 \sigma_B^2 - \sigma_{AB}^2)}$$

$$d_2 = \frac{-[r_B^e - r_{RF}]\sigma_A^2 + [r_A^e - r_{RF}]\sigma_{AB}}{2\lambda(\sigma_A^2 \sigma_B^2 - \sigma_{AB}^2)}$$

To remove $\lambda$,

$$\frac{d_1}{d_2} = \frac{[r_A^e - r_{RF}]\sigma_B^2 - [r_B^e - r_{RF}]\sigma_{AB}}{[r_B^e - r_{RF}]\sigma_A^2 - [r_A^e - r_{RF}]\sigma_{AB}} \qquad (A48)$$

(A48) is the key equation of the mean-variance model, which implies that investors' portfolio decisions will be efficient when the ratio of the shares of the money invested in risky assets A and B is subject to (A48).

When the general indicators $r_A^e$, $r_B^e$, $r_{RF}$, $\sigma_A^2$, $\sigma_B^2$ and $\sigma_{AB}$ are assumed to be $r_{AD}^e$, $r_{HT}^e$, $i - \pi$, $\sigma_{AD}^2$, $\sigma_H^2$ and $\sigma_{AH}$, the optimal portfolio between housing and risky financial assets for each period of time (*t*) will be:

$$\frac{d_1}{d_2} = \frac{g_A A_A}{g_H H} = k^* = \frac{[r_{AD}^e - [i - \pi]]\sigma_H^2 + [r_{HT}^e - [i - \pi]]\sigma_{AH}}{[r_{HT}^e - [i - \pi]]\sigma_{AD}^2 + [r_{AD}^e - [i - \pi]]\sigma_{AH}} \qquad (A49)$$

Equations (A49) is the same as Equation (22).

### Appendix B. Values Used in the Simulations

**Table A1.** Values Used in the Simulations.

| Parameter | Interpretation | Value | Rationale |
|---|---|---|---|
| g_initial | Initial real house price | 40.74 | national level in 2010 |
| ry_initial | Initial real income | 152.03 | national level in 2010 |
| HEtoC | $HE/C$ | 0.203 | national level in 2010 |
| rry_cons | Growth rate of real income | 0.0776 | Sample average |
| std_rry_cons | Standard deviation of rry | 0.0807 | Sample average |
| interest_cons | Nominal interest rate | 0.053 | Sample average (Table 1) |
| inf_cons | Inflation rate | 0.0248 | Sample average |
| dep_cons | Housing depreciation rate | 0.01 | See text |
| theta_cons | CRRA parameter | 5 | See text |
| ra_cons | SSEC growth rate | 0.0577 | sample average |
| std_ra_cons | Standard deviation of ra | 0.0655 | sample average |

Further details and the complete replication of our results are available via our MATLAB simulation codes, which can be provided on request.

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
