# Peer review of "The Effect of Financial Market Factors on House Prices: An Expected Utility Three-Asset Approach"

_axioms, doi:10.3390/axioms11040145_

Round 1
Reviewer 1 Report
My comments and suggestions are in the attachment.

Reviewer 2 Report
This paper explores the role of financial markets as a priced factor in explaining the behaviour of Chinese housing prices/returns. The paper includes the development of a theoretical pricing model and empirical work on China for data from the period from 2001 to 2018. The data shows significant variation in asset return correlations (see the figure on page 11 for a strong illustration).
To what extent do you think your results might be associated with structural breaks associated with the Chinese property market bubble, considering how this may vary across provinces/cities (see Zhi et al 2019) and in response to policy changes (see Jiang and Wang 2021).
References
Jiang, Y., Wang, Y. (2021), Price dynamics of China’s housing market and government intervention, Applied Economics, 53, pp. 1212-1224.
Zhi T., Li Z., Jiang Z., Wei L., Sornette D. (2019), Is there a housing bubble in China? Emerging Markets Review, 39, pp. 120-132.
